# Evolution of cooperation in multichannel games on multiplex networks

**Amit Basak, Supratim Sengupta** *

Department of Physical Sciences, Indian Institute of Science Education and Research Kolkata, Mohanpur Campus, West Bengal, India

* supratim.sen@iiserkol.ac.in

**Data Availability Statement:** Codes used to generate the results in the manuscript and in the electronic supplementary material are available at Zenodo: https://doi.org/10.5281/zenodo.13799987.

**Funding:** This study was partially supported by a MATRICS grant (MTR/2020/000446, 2020-2023),

## Abstract

Humans navigate diverse social relationships and concurrently interact across multiple social contexts. An individual's behavior in one context can influence behavior in other contexts. Different payoffs associated with interactions in the different domains have motivated recent studies of the evolution of cooperation through the analysis of multichannel games where each individual is simultaneously engaged in multiple repeated games. However, previous investigations have ignored the potential role of network structure in each domain and the effect of playing against distinct interacting partners in different domains. Multiplex networks provide a useful framework to represent social interactions between the same set of agents across different social contexts. We investigate the role of multiplex network structure and strategy linking in multichannel games on the spread of cooperative behavior in all layers of the multiplex. We find that multiplex structure along with strategy linking enhances the cooperation rate in all layers of the multiplex compared to a well-mixed population in Prisoners' Dilemma games, provided the network structure is identical across layers. The effectiveness of strategy linking in enhancing cooperation depends on the degree of similarity of the network structure across the layers and perception errors due to imperfect memory. Higher cooperation rates are achieved when the degree of structural overlap of the different layers is sufficiently large, and the probability of perception error is relatively low. Our work reveals how the social network structure in different layers of a multiplex can affect the spread of cooperation by limiting the ability of individuals to link strategies across different social domains.

## Author summary

People hold simultaneous membership in different social networks where they are engaged in interactions with neighbors, some of whom may be common across all networks while others may be unique to a specific social network. The distinct nature of interactions in different social networks can be modeled as games with distinct benefits. How does the nature of the interactions in different social networks as well as the underlying structure of each network affect cooperation levels in the population? Can linking behaviors across different social networks aid in enhancing cooperation levels across all

given to SS by SERB, India. The funders had no role in study design, data collection and analysis, decision to publish, or preparation of the manuscript.

**Competing interests:** The authors have declared that no competing interests exist.

such networks? We use the framework of repeated games on multiplex network-structured populations to address these questions. We find that linking behaviors can be favorable for increasing cooperation levels in all social networks if there is significant structural overlap between the different networks and perception errors in recalling past actions of interacting neighbors, are low.

## 1 Introduction

The conundrum of cooperation stems from the fact that altruistic behavior involves a cost while selfish behavior does not. Despite this disadvantage, cooperation is observed across all biological scales and several mechanisms have been proposed [1–3] to explain how altruistic behavior can be sustained in a population. The Prisoners' Dilemma (PD) game and its variants is often used as a prototypical model to understand evolution of cooperation in well-mixed as well as structured populations since it highlights the tension between short-term personal gain and benefits of mutual cooperation. Two of those mechanisms that are of of significant importance in the context of our work are direct reciprocity [1, 4–22] and network reciprocity [23–31]. The success of direct reciprocity as a strategy for enhancing cooperation is contingent on repeated interactions between a pair of players. In such a scenario, individuals can use conditional strategies [5–7] to retaliate against selfish behavior in future rounds. Previous studies have identified successful strategies in repeated games [9, 10, 13–17] and the conditions under which they can evolve [18–22, 32].

Network reciprocity also plays a significant role in enhancing cooperation levels. In network structured population, interactions of agents are constrained to their immediate network neighborhood. The formation of clusters of cooperators [25, 26] can enable members of the cluster to reap the benefits of mutual cooperation and prevent defectors from over-exploiting them. The spread of cooperation then depends on the benefit-to-cost ratio of cooperative behavior which is crucial in determining whether clusters of cooperators are likely to grow or shrink [23–31]. Simple rules for the evolution of cooperation under different network structures and different update rules, in large [26] and finite [31] populations have been uncovered for PD games.

Certain studies have integrated these two key mechanisms [33–37] of direct reciprocity and network reciprocity in order to understand their impact on the evolution of cooperation. These studies highlight the role of assortative interactions [34, 36], nature of the update process and network connectivity [35] in facilitating the evolution of cooperation. However, most existing studies predominantly focus on a single network structured population, and fail to capture the complexity of human interactions that occur across multiple domains. In reality, people and even organizations frequently engage with others in multiple social contexts concurrently and behavior in one context can influence decisions in other contexts. For instance, academics can socially encounter colleagues in the workplace some of whom also happen to be their collaborators in different research projects. Individuals can belong to different online social networks and different companies are often found to compete in different geographical locations [38]. In each of these cases, regardless of whether the nodes of the network represent individuals or organizations, interaction partners across different domains can either be the same or distinct for each domain. Competition and conflicts in one domain can overtly or subtly affect interactions in the other domains [39].

Despite the complex and unpredictable nature of social environments, humans excel in processing social information and can tailor their behavior accordingly [40]. The benefits yielded and costs incurred by altruistic behavior can vary across social domains. As a result, people

tend to behave differently in domains with different interacting neighbors and can also couple their behaviors across different social domains while dealing with interacting partners that are common across those domains. It is therefore imperative to move beyond well-mixed populations and single layer networks in order to understand the impact of multi-level interactions on the evolution of cooperation. Multiplex networks provide the ideal framework [40–43] for addressing these issues [44–52]. The different layers of a 2-layer multiplex network have been used [53, 54] to distinguish between the social network where the game is played (called the interaction network) and the strategy update network that is used to update strategy through imitation. The multiplex network reduces to a single network in the limit where the interaction and strategy update networks are identical. However, an asymmetry between the interaction and strategy update network was shown [46, 52] to affect cooperation levels in the population.

Several groups have analyzed evolutionary games on multiplex structured networks and highlighted the benefits afforded by multiple layers [45, 48], asynchronous strategy update [50] and structural similarity between layers [49] in promoting cooperation in such systems. Gomez-Gardenes *et al.* [45] analyzed the effect of multiple layers, using a *single* PD game that was played in all layers, and found that cooperation can be sustained even for a high temptation to defect, with the resilience of cooperation increasing with the number of layers of the multiplex. Pereda [48] considered a duplex network where one layer represents the strategies of players involved in a PD game with their neighbors and another layer represents the extent to which the players are vigilant of the actions of neighbors playing the PD game. An individual's temptation to defect was regulated by the extent to which her actions were scrutinized by her neighbors. By coupling both strategies and vigilant behavior across layers and updating both over time using a pairwise payoff comparison rule, they show how cooperation levels are affected by the duplex network structure as well as the extent of structural similarity between the layers. Allen et al. [50] compared the effect of synchronous and asynchronous updating using a PGG game on a duplex network and found that asynchronous update of strategies across layers promotes cooperation for lower values of synergy factor. Battiston *et al.* [49] were the first to systematically study the impact of the structural overlap of the different layers of a multiplex network on the evolution of cooperation, using a PGG. They showed that the dominance of cooperators depends on the significant structural overlap between the layers in addition to a high synergy factor associated with the PGG game in at least one layer of the multiplex.

In all these models, different layers of the multiplex were coupled through total payoff (calculated by adding the payoffs from interactions with all neighbors across all layers of the multiplex) and the strategy update process (where an individual in one layer can imitate the strategy of a neighbor in another layer). However, by focusing on one-shot games, these previous studies on multiplex networks disregarded the interplay between an individual's behavior in one layer and its potential influence in another layer. In contrast, using the framework of direct reciprocity, we consider repeated games across layers, where individuals can link their behaviors to retaliate against a common co-player's defection in one layer by defecting in another layer in future rounds.

A recent study by Donahue *et al.* [55] have highlighted the positive impact of linking strategies across multiple games on the evolution of cooperation. They found that coordinating strategies across multiple games can lead to enhanced cooperation levels across all games. However, such investigations have ignored the potential role of network structure, focusing only on analyzing the effects of linking strategies across multiple games occurring in a single well-mixed population.

In this paper, we use a multiplex network to describe social interactions that occur across multiple domains. A population of individuals simultaneously engage in multiple repeated games, each occurring on a different layer of the multiplex. The different games are distinguished by distinct benefit-to-cost ratio for altruistic behavior. The network structure on each

layer of the multiplex can either be identical to or distinct from other layers with the deviation from complete structural overlap of the layers being determined by the average edge overlap $\mathcal{O}$ between the two layers of the multiplex. As a consequence, each individual can have a set of connections with neighbors in one layer that are distinct from her connections in another layer ($0 \leq \mathcal{O} < 1$). When there exists complete structural overlap between the layers ($\mathcal{O} = 1$), each individual has the same set of connections across all layers of the multiplex. Individuals can link their strategies from multiple games against their common interaction partners across layers (represented by black edges in Fig 1a) to induce cooperative behaviors in those social contexts where cooperation is relatively costly. However, an agent treats each game independently while playing against unique interaction partners (represented by red edges in Fig 1a) in different layers. Each individual's payoffs arising from interactions in each layer are aggregated across all layers. In our model, layers of multiplex are not only coupled through payoffs; individuals can also link their behavior against common neighbors across layers. By studying repeated PD games on a multiplex, we wish to investigate the impact of multiplex structure, degree of structural overlap between layers, variable benefits of cooperation across different layers on the emergent levels of cooperation. Our model extends beyond analyzing the role of multiplex network structures in multichannel games. Using a theoretical framework, we also explore the effect of perception errors in recalling past actions, due to imperfect memory, in multiple repeated games with both unlinked and linked strategies.

Our results indicate that the outcome of evolutionary dynamics in multiplex network structured populations is often in stark contrast to that in well-mixed populations. This is observed both in the absence and presence of strategy linking. We find that strategy linking against common neighbors is only effective in promoting cooperation when the degree of edge overlap between layers is substantial. Our results underscore the importance of population structure and imperfect memory on the evolution of cooperation in multiple repeated games.

The paper is structured as follows: Sec. 2 elaborates on the methodologies employed for calculating payoff and cooperation rate in the context of multiple repeated games on a multiplex network. In Sec. 3, we present all the main results of the paper and finally conclude with a discussion of the key results and possible future directions of research in Sec. 4.

## 2 Methods

We consider a population of individuals engaging in distinct games in different layers of an $m$-layer multiplex network, with each layer having an identical number ($N$) of nodes. Each individual is represented by a node that can potentially have different neighbors in different layers. Each layer $\alpha$ of the multiplex is represented by an adjacency matrix $A^\alpha = \{A_{ij}^\alpha\}$, where $A_{ij}^\alpha = 1$ when individuals $i$ and $j$ are connected in layer $\alpha$, and $A_{ij}^\alpha = 0$ otherwise. A vector of adjacency matrices characterizes a multiplex network of $m$ layers, $\mathbf{A} = \{A^1, A^2, \ldots, A^m\}$.

### 2.1 Key properties of multiplex networks

The degree of a node $i$ in a specific layer $\alpha$ of a multiplex is calculated as, $k_i^\alpha = \sum_{j \neq i} A_{ij}^\alpha$, from which it follows that $0 \leq k_i^\alpha \leq N - 1, \forall i, \forall \alpha$. Thus, the multiplex degree of node $i$ is a vector $k_i = \{k_i^1, \ldots, k_i^m\}$. If, $k_i^\alpha = N - 1 \ \forall i, \forall \alpha$, then each layer of the multiplex network is a complete graph and the multiplex network reduces to a single well-mixed population.

A key feature of a multiplex network is that a pair of nodes can be connected in one layer but not in others. As illustrated in Fig 1 which shows a multiplex network with two layers, a node may have a different set of neighbors in layer 2 compared to layer 1. For example, the node $j$ acts as a common neighbor of node $i$ in both layer 1 and layer 2 (indicated by black

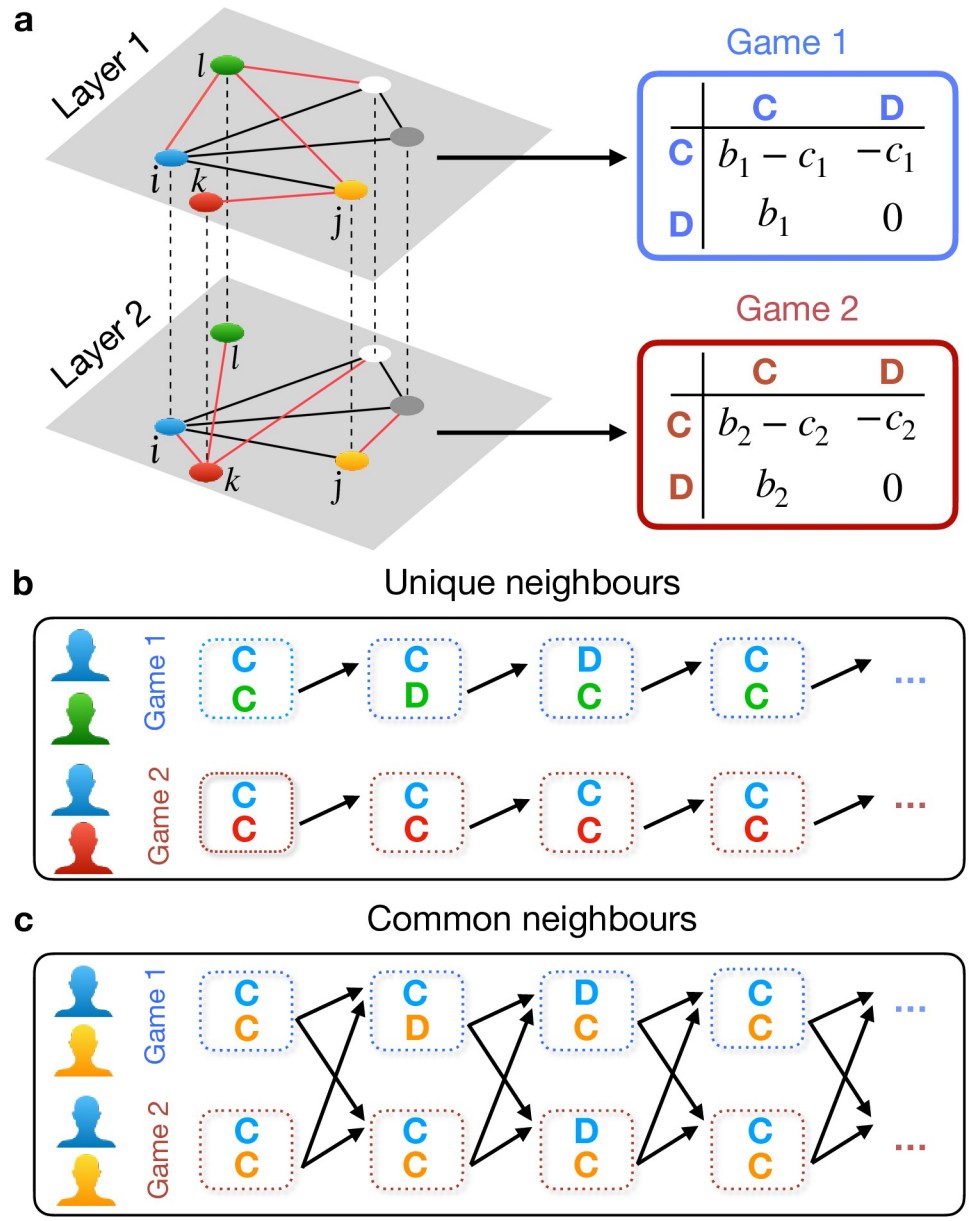

**Fig 1. Schematic of multichannel games on a two-layer multiplex network.** (a) Two distinct donation games, each with different benefits and costs of cooperation, are played across the layers. Each individual is represented by a node in layer one, with dotted lines indicating connections to their counterparts in the other layer. Red edges denote unique connections, while black edges signify common connections between pairs of nodes across both layers. (b) Players treat each game independently when interacting with unique neighbors in each layer. A player (blue) responds to a unique neighbor's (green in layer 1, red in layer 2) previous action within the same game and layer. This is indicated by an arrow from the opponent's action in the previous round to the focal player's response (action) in the current round. (c) When interacting with a common partner, a player (blue) can link the two games, reacting to the opponent's (yellow) defection in one game by defecting in both games. This linkage is assumed to occur only in interactions with common neighbors. This is indicated by two arrows pointing from the common opponent's actions in the last round in both games to the player's actions in the current round.

edges). However, node $i$ is exclusively connected to node $k$ in layer 2 (illustrated by a red edge), without a corresponding connection in layer 1.

To quantify the degree of similarity between layers, we consider the edge overlap between a pair of nodes $i$ and $j$, defined as, $o_{ij} = \frac{1}{m}\sum_{\alpha=1}^{m} A_{ij}^{\alpha}$, where $m$ is the number of layers of a multiplex. $o_{ij}$ is 1 if the edge $i - j$ is common across all layers. The number of common neighbors $O_i^{CN}$ across a pair of layers $\alpha, \beta$ and unique neighbors $O_i^{\alpha UN}$ of a node $i$ in layer $\alpha$ are respectively defined as

$$
\begin{aligned}
O_i^{CN} &= \sum_{j\neq i} A_{ij}^{\alpha} A_{ij}^{\beta} \\
O_i^{\alpha UN} &= k_i^{\alpha} - \sum_{j\neq i} A_{ij}^{\alpha} A_{ij}^{\beta}
\end{aligned}
\tag{1}
$$

The average (avg.) fraction of common neighbors (i.e. edge overlap) across a pair of layers $\alpha, \beta$, is given by:

$$
\mathcal{O}^{\alpha,\beta} = \frac{\sum_{i,j>i} A_{ij}^{\alpha} A_{ij}^{\beta}}{\sum_{i,j>i} A_{ij}^{\alpha} + A_{ij}^{\beta} - A_{ij}^{\alpha} A_{ij}^{\beta}}
\tag{2}
$$

For all pairs of $m$ layers of a multiplex [56], the above quantity can be generalized to $\mathcal{O} = \frac{2}{m(m-1)}\sum_{\alpha,\beta>\alpha}^{m} \mathcal{O}^{\alpha,\beta}$. When all layers of a multiplex are the same (i.e. have identical topologies and edges), the average edge overlap of a multiplex becomes maximum, $\mathcal{O} = 1$. Similarly, when all layers are completely distinct from each other (each node has different connections across layers), then edge overlap is minimum, $\mathcal{O} = 0$. In our model the average fraction of *unique* neighbors is denoted by $1 - \mathcal{O}$. The algorithm used to create a multiplex network with desired edge-overlap between the layers is described in Sec.S1 of the S1 Text.

## 2.2 Strategy employed against common and unique neighbors

In our framework, two distinct games, characterized by distinct payoff matrices, are being played in the two layers of the multiplex network. Each individual engages in both games repeatedly and simultaneously with their respective neighbors in each layer, taking a decision to Cooperate (C) or Defect (D) in each round. Although the formalism described here is valid for any value of $m$, we have used $m = 2$ throughout this paper. Players use reactive strategies, where behavior depends only on the co-player's previous actions. In our model, agents employ distinct strategies based on the nature (common or unique) of their neighbors across network layers. When interacting with common neighbors, the strategy is based on the collective actions of the co-player in the last round in *both* layers. Such strategies have been called [55] multi-game linked reactive strategies. Specifically, a player playing against a common neighbor (CN) in the two layers of a multiplex network, the player's strategy takes the form of a 10 tuple,

$$
\mathbf{p}_{CN} = (p_{00}^1, p_{CC}^1, p_{CD}^1, p_{DC}^1, p_{DD}^1; p_{00}^2, p_{CC}^2, p_{CD}^2, p_{DC}^2, p_{DD}^2)
\tag{3}
$$

$p_{a_1,a_2}^{\alpha}$ is the player's probability to cooperate in layer $\alpha$, depending on the co-player's previous actions $a_1$ and $a_2$ in layer 1 and layer 2 respectively. The first five components of the strategy vector signify the player's strategy in layer 1, while the last five components denote her strategy in layer 2. The first entry for each layer $p_{00}^{\alpha}$ is the initial probability of cooperating against a common neighbor in layer $\alpha$.

When playing against a unique neighbor (UN) in a specific layer $\alpha$, players are unable to condition their behavior on the opponent's action in other layers, and strategy against such neighbors depends only on the co-player's last-round action in the layer $\alpha$. Such a strategy has

been called [55] a multi-game unlinked reactive strategy and is defined as

$$\mathbf{P}^\alpha_{UN} = (p^\alpha_0, p^\alpha_C, p^\alpha_D) \tag{4}$$

We note that the set of unique neighbor strategies is a strict subset of the common neighbor strategies (they correspond to those $\mathbf{p}_{CN}$ for which $p^\alpha_{a_1,a_2} = p^\alpha_{a_\alpha}$ for all layer $\alpha$, and all actions $a_1$ and $a_2$). We consider finitely repeated games with $w$ as the probability of another round of games. For infinitely repeated prisoner's dilemma game between the focal player and each of her neighbors in both layers, the first round probability to cooperate becomes irrelevant.

## 2.3 Payoff calculation

A player's payoff in each game $\alpha$ (occurring in layer $\alpha$) is either $R_\alpha$, $S_\alpha$, $T_\alpha$, or $P_\alpha$, depending on the player's and the co-player's action. In general, any $2 \times 2$ game can be classified on the basis of the dilemma strength parameters $D'_g = \frac{T_\alpha - R_\alpha}{R_\alpha - P_\alpha}$ (Gamble Intending Dilemma (GID)) and $D'_r = \frac{P_\alpha - S_\alpha}{R_\alpha - P_\alpha}$ (Risk Averting Dilemma (RAD)) [57–60]. The Prisoners' Dilemma (PD), Chicken or Snowdrift (SD), Stag-Hunt (SH) and Trivial or Harmony (H) games correspond to $D'_r > 0$ and $D'_g > 0$, $D'_r < 0$ and $D'_g > 0$, $D'_r > 0$ and $D'_g < 0$, $D'_r < 0$ and $D'_g < 0$ respectively. In this paper our primary focus is on the Donation game which is a version of the PD game for which $D'_g = D'_r$. In such a game, the payoff matrix elements become $R_\alpha = b_\alpha - c_\alpha$, $S_\alpha = -c_\alpha$, $T_\alpha = b_\alpha$, and $P_\alpha = 0$, where $b_\alpha$ and $c_\alpha$ are the benefit and cost of cooperation in game $\alpha$.

The expected pairwise payoff for player $i$ (or $j$) in the infinitely repeated game, when playing against a neighbor $j$ (or $i$) in layer $\alpha$ are respectively given by

$$\pi^\alpha_{ij} = v^\alpha_{CC} R_\alpha + v^\alpha_{CD} S_\alpha + v^\alpha_{DC} T_\alpha + v^\alpha_{DD} P_\alpha$$
$$\pi^\alpha_{ji} = v^\alpha_{CC} R_\alpha + v^\alpha_{CD} T_\alpha + v^\alpha_{DC} S_\alpha + v^\alpha_{DD} P_\alpha \tag{5}$$

where $v^\alpha_{CC}$, $v^\alpha_{CD}$, $v^\alpha_{DC}$, and $v^\alpha_{DD}$ represent the equilibrium probability of finding the game in layer $\alpha$ in the state $a\tilde{a}$ where $a, \tilde{a}$ correspond to the actions of player 1 and player 2 respectively in layer $\alpha$ (refer to Sec.S2 of S1 Text for detailed calculation of $v^\alpha_{a\tilde{a}}$).

The accumulated average payoff of player $i$ in layer $\alpha$ due to pairwise interactions with all neighbors (including both common and unique neighbors) is,

$$\pi^\alpha_i = \frac{1}{k^\alpha_i} \left( \sum_{j \neq i} \delta_{1,o_{ij}} \pi^\alpha_{ij} + \sum_{j \neq i} (1 - \delta_{1,o_{ij}}) A^\alpha_{ij} \bar{\pi}^\alpha_{ij} \right)$$
$$= \pi^{CN}_i + \pi^{UN}_i \tag{6}$$

Where, the Kronecker delta term equals 1 when node $j$ is a common neighbor of $i$ across all layers (i.e. $o_{ij} = 1$) and is 0 otherwise. $\pi^\alpha_{ij}$ ($\bar{\pi}^\alpha_{ij}$) is pairwise payoff of $i$ against a common (unique) neighbor $j$ in layer $\alpha$ using strategy $\mathbf{p}_{CN}$ ($\mathbf{p}^\alpha_{UN}$) respectively.

The total expected payoff for player $i$ as a consequence of her interactions with all neighbors across all layers of the multiplex network is

$$\pi_i = \sum_{\alpha=1}^m \pi^\alpha_i \tag{7}$$

## 2.4 Cooperation rate calculation

Player $i$'s (or alternatively, player $j$'s) average cooperation rate when playing against player $j$ (or alternatively, player $i$) in layer $\alpha$ respectively is,

$$\gamma_{ij}^{\alpha} = v_{CC}^{\alpha} + v_{CD}^{\alpha}$$
$$\gamma_{ji}^{\alpha} = v_{CC}^{\alpha} + v_{DC}^{\alpha}$$

(8)

The average cooperation rate of player $i$ against common and unique neighbors in layer $\alpha$ can be calculated separately by averaging her cooperation rate against each type of neighbor respectively, in that specific layer.

$$\zeta_{i,CN}^{\alpha} = \frac{1}{O_i^{CN}} \sum_{j \neq i} \delta_{1,o_{ij}} \gamma_{ij}^{\alpha}$$

$$\bar{\zeta}_{i,UN}^{\alpha} = \frac{1}{O_i^{\alpha UN}} \sum_{j \neq i} (1 - \delta_{1,o_{ij}}) A_{ij}^{\alpha} \bar{\gamma}_{ij}^{\alpha}$$

(9)

Where, $O_i^{CN}$ and $O_i^{\alpha UN}$ is number of common and unique neighbors (in layer $\alpha$) respectively of player $i$ and $\gamma_{ij}^{\alpha}$ is cooperation rate of player $i$ against common neighbor $j$ in layer $\alpha$. Similarly, $\bar{\gamma}_{ij}^{\alpha}$ is cooperation rate of $i$ against a unique neighbor $j$ in layer $\alpha$, calculated using the unlinked reactive strategy ($p_{a_\alpha}^{\alpha}$). (For a detailed calculation of $\gamma_{ij}^{\alpha}$ and $\bar{\gamma}_{ij}^{\alpha}$ refer to Sec.S2 of S1 Text). Similarly, an individual $i$'s average cooperation rate in layer $\alpha$ against all of her neighbors is,

$$\zeta_i^{\alpha} = \frac{1}{k_i^{\alpha}} \left( \sum_{j \neq i} \delta_{1,o_{ij}} \gamma_{ij}^{\alpha} + \sum_{j \neq i} (1 - \delta_{1,o_{ij}}) A_{ij}^{\alpha} \bar{\gamma}_{ij}^{\alpha} \right)$$

(10)

At each time step, the population's average linked and unlinked cooperation rate in game $\alpha$ (occurring in layer $\alpha$) against common neighbor and unique neighbors respectively is calculated by averaging over the whole population for each type separately.

$$\zeta_{CN}^{\alpha} = \frac{1}{N} \sum_{i=1}^{N} \zeta_{i,CN}^{\alpha}, \qquad \bar{\zeta}_{UN}^{\alpha} = \frac{1}{N} \sum_{i=1}^{N} \bar{\zeta}_{i,UN}^{\alpha}$$

(11)

$\zeta_{CN}^{\alpha}$ is population's average linked cooperation rate against common neighbors, calculated using linked reactive strategies ($p_{a_1,a_2}^{\alpha}$) and similarly, $\bar{\zeta}_{UN}^{\alpha}$ is population's average unlinked cooperation rate against unique neighbors in layer $\alpha$, calculated using unlinked reactive strategies ($p_{a_\alpha}^{\alpha}$).

At each time step, the population's average cooperation rate is calculated as,

$$\zeta^{\alpha} = \frac{1}{N} \sum_{i=1}^{N} \zeta_i^{\alpha}$$

(12)

For all future instances, we will refer to the population's average cooperation rate and average linked (unlinked) cooperation rate as cooperation rate and linked (unlinked) cooperation rate only.

## 2.5 Categorization of outcomes

Simulation outcomes are classified on the basis of the population's cooperation rates [55].

Full cooperation: $\zeta^{\alpha} \geq 0.8 \; \forall \; \alpha$

Only layer $\alpha$ cooperation: $\zeta^\alpha \geq 0.8$ and $\zeta^\beta \leq 0.2 \; \forall \; \beta \neq \alpha$

No cooperation: $\zeta^\alpha \leq 0.2 \; \forall \; \alpha$

## 2.6 Strategy update

At each time step of the simulation, the population updates their strategy according to two different updating methods. They either explore random strategies within their respective strategy spaces with probability $\mu$, or with probability $1 - \mu$, imitate a randomly chosen neighbor's strategy based on the neighbor's perceived success [61, 62]. The imitation process followed the pairwise comparison rule [61, 62], where an individual $A$, would imitate the strategy of another individual, $B$, with a probability of $p = (1 + \exp(-s(f_B - f_A)))^{-1}$. Here, $f_A$ ($f_B$) represents the payoffs of individual $A(B)$, and $s$ denotes the intensity of selection. Since each individual is equipped with both multi-game linked and unlinked reactive strategies to interact with their common and unique neighbors respectively, both strategies of the focal player need to be updated. Consequently, during the imitation process, player $A$ will adopt both strategies of player $B$ according to the pairwise comparison rule. We have also explored two different update schemes.

**i. Independent strategy update**: At each time step, a single network layer was randomly selected, and all individuals within that network were given the opportunity to update their strategies.

**ii. Simultaneous strategy update**: At each time step, all individuals across all layers update their strategy. If a common neighbor is selected for imitation by the focal player in any one layer, the strategy of the focal player is updated across all layers. If an unique neighbor is selected for imitation in any one layer, the strategy of the focal player is updated only in the corresponding layer.

## 3 Results

We analyze the effect of complex network structure on the cooperation rate in multi-channel games on multiplex networks by considering different types of complex networks, such as random-regular networks (RRN) and scale-free networks (SFN), having identical topologies and identical connections across all layers of the multiplex network. Such an analysis allows us to understand how the underlying network structure affects cooperation levels in contrast to well-mixed populations represented by a complete graph. We then analyze 2-layer multiplex networks having identical topologies but distinct connections across layers such that the same node of the network in different layers can have a different set of connections. This allows us to understand how the degree of edge overlap between identical nodes in the different layers affect cooperation levels in multichannel games where strategies can be linked only for interactions with those neighbors of a node that are common across both layers.

## 3.1 Evolutionary dynamics in multiplex networks having identical topologies and identical connections across layers

We work with a multiplex network having two layers and both layers have RRN topology with $N$ nodes. Each node is connected at random to $k$ other nodes. The $k$ neighbors of each node are common across both layers of the multiplex network ($\mathcal{O} = 1$) as a consequence of which the behavior of each individual in the network can be linked across all layers of the multiplex. This enables each individual to condition her behavior on the opponent's actions in *all* games. For comparison, we also consider a scenario where individuals treat each game independently, conditioning their behavior on the opponent's actions in that specific game. Since each node has the same set of connections in both layers, our model of a 2-layer network topology

effectively reduces to a single network topology with maximal edge overlap. However, the two layers of the network can still be distinguished from each other if the two games played on the two networks are distinct, i.e. have different payoff matrices. This scenario generalizes the work of Donahue et al. [55], which considered a mixed population which arises as a limiting case of our model when networks in all layers are described by a complete graph i.e. a graph with $k = N - 1$.

We assume (unless specified otherwise) that the game played in layer 1 has a higher benefit of cooperation, such that $b_1 > b_2$. Despite the smaller benefit in the second game, very high cooperation evolves in both games, reaching 94.8% in the first game and 80.6% in the second, when individuals treat each game independently (Fig 2a–2d) and interact with just four neighbors ($k = 4$). As the degree of the multiplex network increases, high cooperation rate persists with unlinked strategies only in game 1; but the cooperation rate steadily decreases in game 2 (Fig 2a). However, when individuals use linked strategies, higher cooperation rate is observed in *both games*, even for high degree (Fig 2e). Fig 2b and 2f shows the temporal evolution of cooperation rates for unlinked and linked strategies, respectively, when each layer in the multiplex has a degree $k = 4$. In order to better understand these results, we plot the abundance of different evolved cooperation scenarios (see Sec. 2.5) in the population. Contrary to Donahue *et al.* [55], we find that the full cooperation scenario is the most likely evolutionary outcome for both unlinked (Fig 2c) and linked (Fig 2g) cases when individuals interact with only four neighbors, as opposed to a well-mixed population where everyone interacts with everyone else ($k = N - 1$). These results suggest that the multiplex network structure itself can promote cooperation in both layers when the connectivity of the network is low, irrespective of strategy linking. However, for high network connectivity (i.e. large $k$), linked strategies are more effective in promoting cooperation in both layers of the multiplex compared to unlinked strategies.

To understand how a network structured population with degree $k = 4$ facilitate high cooperation in both games, we examine the population's average strategy at the end of simulation. The average strategy in any layer $\alpha$ is obtained by averaging $p^{\alpha}_{a_1, a_2}$ in the linked case (or $p^{\alpha}_{a_{\alpha}}$ in the unlinked case) over the population and then further averaging the result over multiple trials. In the unlinked case (Fig 2d), the average strategy in layer 1, resembles Generous Tit-for-Tat (GTFT), where individuals reciprocate altruistic behavior with a high probability ($p^1_C > 0.95$) in layer 1 and still cooperate with a moderate probability ($p^1_D \approx 0.4$) in response to a co-player's defection. However, the evolved strategies in layer 2 are less cooperative ($p^2_C < p^1_C, p^2_D < p^1_D$), leading to a lower cooperation rate in game 2. In the linked case (Fig 2h), on average, individuals cooperate with high probability across all layers in response to cooperation in both games, but lower their cooperation probability in both games if the co-player defects in either. Under the independent strategy update scheme, we observed that the evolved strategy in the population cooperates with high probability only when the co-player either cooperates or defects in both games in the previous round. For simulations using deterministic strategies, the evolved strategy can be represented as, $\mathbf{p} = (1, 0, 0, 1; 1, 0, 0, 1)$. This strategy ought to be vulnerable to invasion by ALLD strategy, which defects in both games. However, for an ALLD strategy to emerge, a single-layer defector strategy, e.g. $\mathbf{p}' = (0, 0, 0, 0; 1, 0, 0, 1)$ or $(1, 0, 0, 1; 0, 0, 0, 0)$ has to first evolve in the resident population of the evolved cooperative strategy $\mathbf{p}$ under independent strategy update. But, when such a single-layer defector strategy starts defecting in one game, it is punished in both due to the low $p_{CD}$ and $p_{DC}$ values of the evolved strategy, $\mathbf{p}$ in both games. For an infinitely repeated multichannel game between $\mathbf{p}$ and $\mathbf{p}'$, in the stationary distribution they will be in the *DDDC* or *DDCD* state of the Markov chain with a probability close to 1. As a result, the evolved cooperative strategy gets a payoff $b_2$ or $b_1$, whereas the single-layer defector strategy gets $-c_2$ or $-c_1$. Thus, the single-layer defector

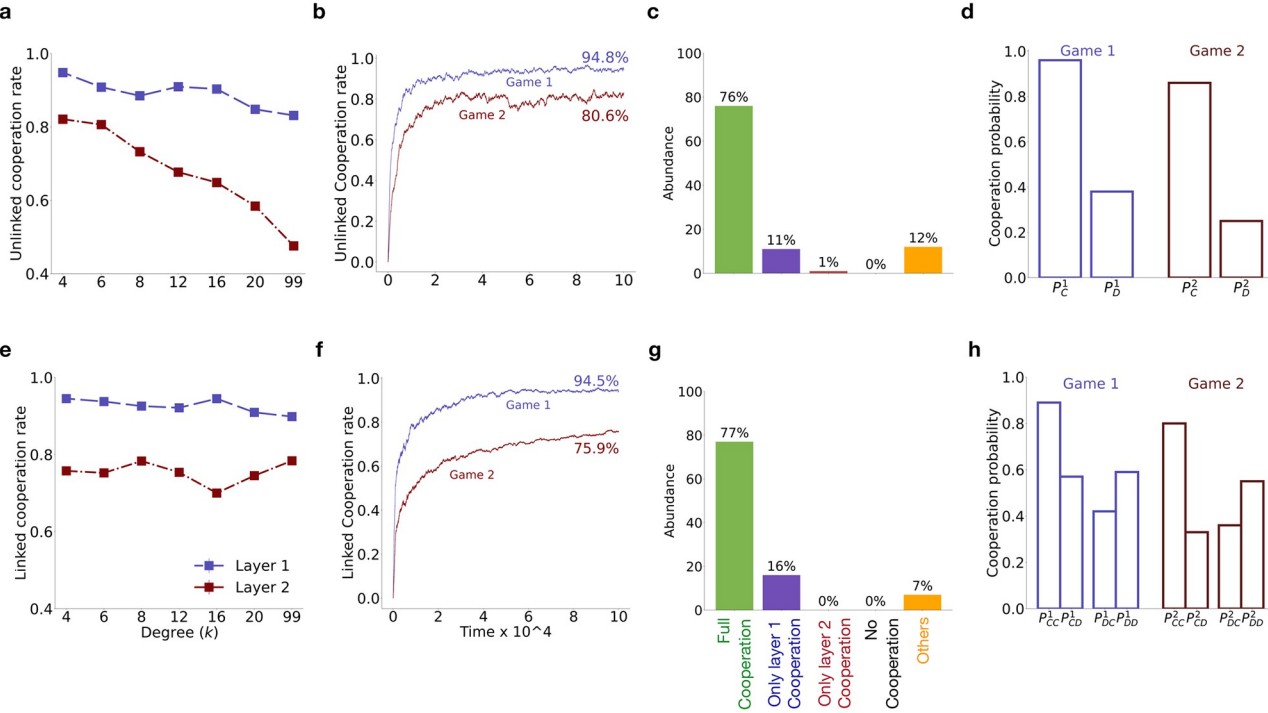

**Fig 2. Impact of network connectivity.** Evolution of cooperation in multiplex structured population using unlinked **(a-d)** and linked **(e-h)** strategies. **(a,e)** shows the variation of the cooperation rate with degree ($k^1 = k^2 = k$); **(b,f)** the time evolution of the cooperation rate for $k = 4$; **(c,g)** relative abundance of different cooperation scenarios across multiple simulations for $k = 4$; **(d,h)** the average strategy used for $k = 4$; when all individuals use unlinked and linked strategies respectively. The results were obtained by averaging over 200 simulations using the independent strategy update rule and RRN topology for each layer of the multiplex network. Other parameters used: $N = 100$, $b_1 = 5$, $b_2 = 3$, $c_1 = c_2 = 1$, $\mu = 0.001$, $w = 1$, and $s = 2$.

strategy is at a disadvantage in a population of strategies defined by **p**, making the emergence of the ALLD strategy very unlikely. Hence, evolved cooperative strategies such as **p** remain robust under the independent strategy update scheme.

When we simulated the evolutionary dynamics using the simultaneous strategy update scheme, a similar pattern was observed in the evolution of the linked cooperation rate in both layers (Fig 3a) and the abundance of different cooperation scenarios (Fig 3b). However, the average strategy in the full cooperation scenario (see Fig 3c) are mutually cooperative with a high probability ($p_{CC}^1 = p_{CC}^1 \approx 1$) and tend to cooperate with a similar probability, comparable to what is expected by chance, when opponent defects in either or both games.

In order to understand if the persistently higher linked cooperation rate is robust to changes in population size as well as network topology of the layers, we carried out simulations for networks of varying sizes as well as different topologies for two different update schemes and finitely repeated games in both layers. Our results generally indicate that a higher linked cooperation rate is observed in both Game 1 Fig 4a and 4d and Game 2 Fig 4b and 4e in structured populations (multiplex created with RRN and SFN), across all sizes, when compared with the mixed population scenario. The effect of network structure on the linked cooperation rate for both games is more pronounced for large population size when the simultaneous strategy-update scheme is used. As shown in Fig 4c and 4f full cooperation scenario is more prevalent in structured populations as compared to mixed populations. We find that direct reciprocity along with strategy linking in multichannel games, even when acting together, may not be sufficient to ensure high levels of cooperation in well-mixed populations when the population

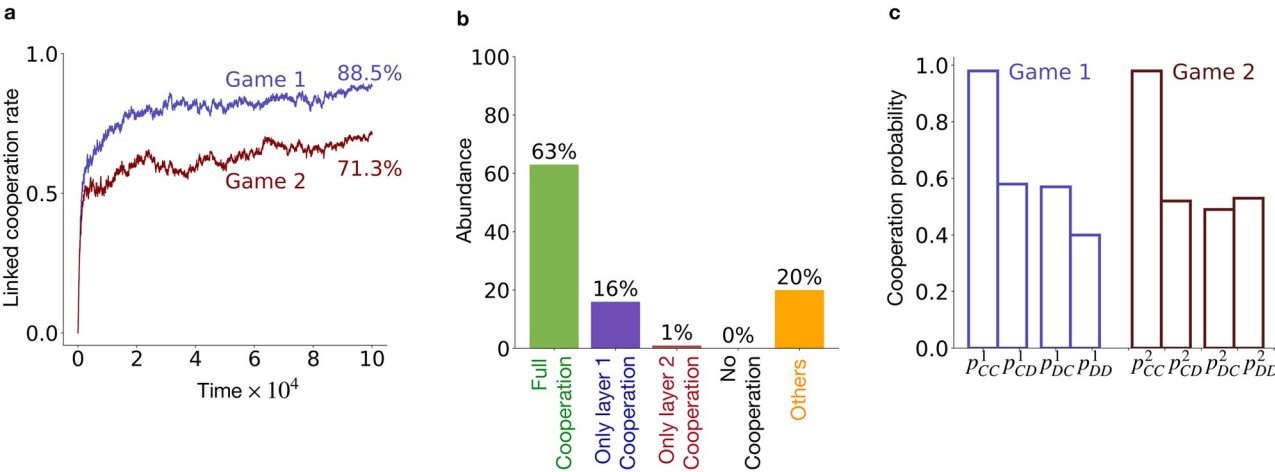

**Fig 3. Evolutionary dynamics under the simultaneous strategy update scheme.** (a) Evolution of linked cooperation rate in network layers 1 and 2. (b) Relative abundance of different evolved cooperation scenarios obtained across 100 independent simulations.(c) Population's average strategy in the full cooperation scenario. Each bar represents population's average of $p_{a_1,a_2}^\alpha$ in the respective layer, averaged over 100 independent realisation where full cooperation scenarios evolved. The 2-layer multiplex network had identical RRN topology with $k = 4$ in each layer. All other parameter values are same as Fig 2.

size is large. However, the presence of a multiplex network structure in conjunction with strategy linking can significantly enhance cooperation in multichannel games (see Fig 4d–4f) even for large populations.

We have also verified that our results are in agreement with that of Donahue et al. [55] when the underlying network is a complete graph in both layers for the case where $N = 50$ and $\mu = 0.001$ which corresponds to the low mutation limit as investigated in [55].

## 3.2 Evolutionary dynamics of a reduced set of strategies

We carried out an analysis using reduced strategy sets in order to better understand how the structure of the underlying multiplex network in multichannel games affects the nature of the emergent linked strategy.

For the purpose of our analysis, we consider a three-strategy system, consisting of Linked GTFT (LGTFT) = (1, 1, $q$, $q$, $q$;1, 1, $q$, $q$, $q$), ALLD = (0,0,0,0,0; 0,0,0,0,0) and ALLC = (1,1,1,1,1; 1,1,1,1,1). The LGTFT strategy considered here is an approximation of the average strategy that evolved in the full cooperation scenario ($\zeta^\alpha \geq 0.8\ \forall \alpha$), for the simultaneous strategy update scheme (Fig 3c). This strategy starts by cooperating and continues to cooperate in all games as long as the opponent also cooperates. However, if the opponent defects in any of the two games, LGTFT immediately lowers her cooperation probability in all games to $q\left(< 1 - \frac{c_1+c_2}{b_1+b_2}\right)$. Due to the similarity of this strategy with the well-studied Generous Tit-for-Tat (GTFT) [6] that emerges in the context of single repeated games, we have named this strategy LGTFT. In the limit $q = 0$, the LGTFT strategy reduces to the Linked TFT (LTFT) strategy.

We first analyze a 2-strategy system to identify the conditions under which the LGTFT strategy can invade a population of ALLD players in both well-mixed and structured populations. The conditions, $\frac{b_1+b_2}{c_1+c_2} > \frac{3-2(1-q)w}{(1-q)w}$ and $\frac{b_1+b_2}{c_1+c_2} > \frac{3-2(1-q)w}{(1-q)w} - \frac{3(1-(1-q)w)}{(k+1)(1-q)w}$ respectively, translate to lower bounds on the benefit ($b_1$) in game 1 (see Sec.S3.1 of S1 Text for details). We then carry out simulations, in the absence of mutations, to obtain the fixation probability of the LGTFT

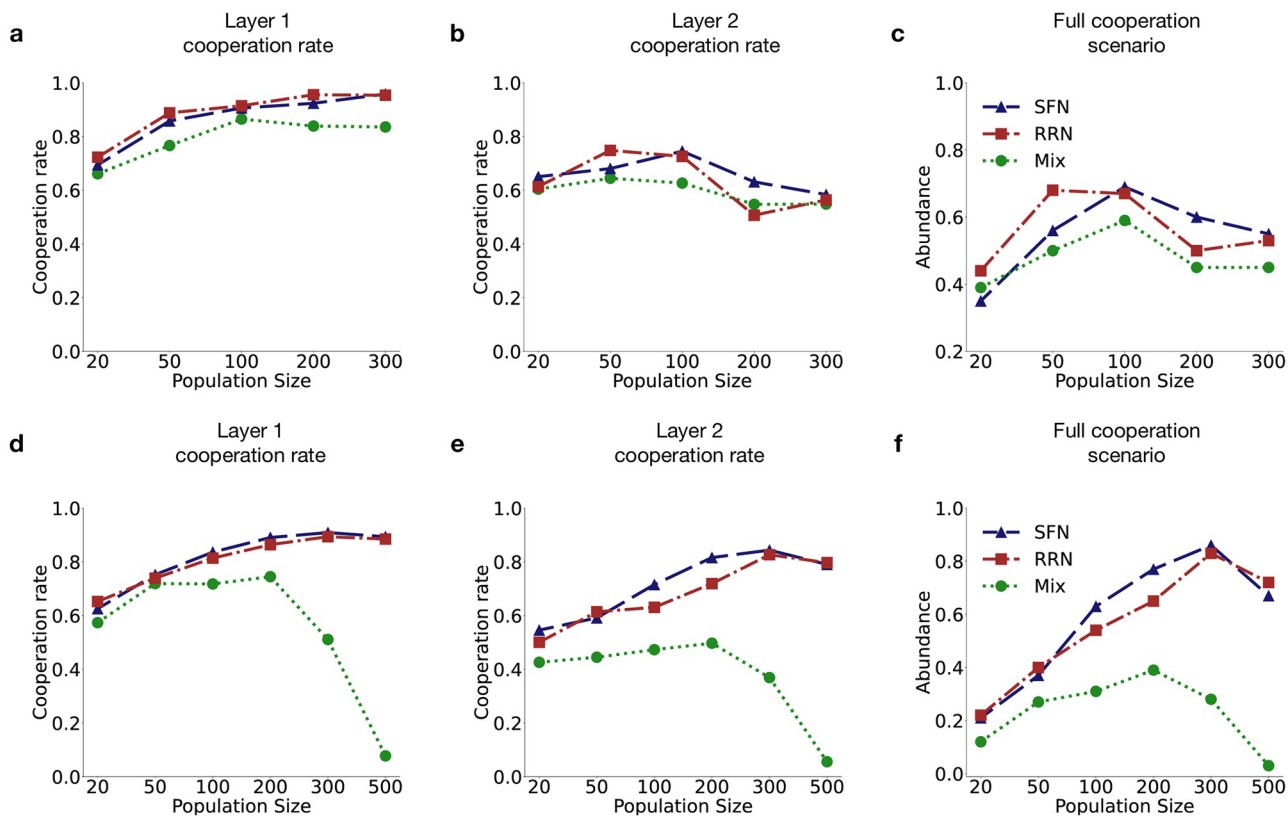

**Fig 4. Comparison between evolved linked cooperation rates for different topologies and population sizes.** Each data point depicts the cooperation rate of population in Layer 1 (**a,d**), in Layer 2 (**b,e**) at the end of the simulation and (**c,f**) the fractional abundance of full cooperation scenarios across 100 independent runs. Panels **a-c** correspond to the independent strategy update scheme and panels **d-f** correspond to simultaneous strategy update scheme respectively. $w = 0.95$ was used. For SFN networks, average degree $\bar{k} = 4$. Other parameters values used are same as in Fig 2.

strategy in a resident population of ALLD for different values of the benefit in game 1 ($b_1$) while keeping the benefit in game 2 ($b_2$) fixed (Fig 5). In a *large but finite* population, we observe that the critical benefit required in game 1 for the fixation probability of LGTFT to exceed the neutral fixation probability of $1/N$ is significantly lower when interactions of each node are constrained to a few connected neighbors (for example, when $k = 3, 5$) compared to the well-mixed case (where $k = N − 1$) and agree well with our theoretical predictions (see Sec. S3.1 of S1 Text for details).

In a system with three strategies, the abundance of both LGTFT and ALLD strategies exceeds 1/3 in both well-mixed and structured populations in the weak selection limit when $2q^* − 1 < q < q^*$, where $q^* = \frac{(b_1+b_2)−(c_1+c_2)}{(b_1+b_2)+(c_1+c_2)}$. However, for LGTFT to dominate over both ALLD and ALLC, the conditions $q < 2q^* − 1$ and $\frac{b_1+b_2}{c_1+c_2} \geq 3$ need to be simultaneously satisfied (see section S3.2 of S1 Text for details). Numerical simulations of the competition between these three strategies provides additional insights into the dynamics for arbitrary selection strengths. Fig 6 shows a heat map depicting the relative abundance of the three strategies in the low mutation limit. Each point in the simplex $S_3$ represents the long-term average frequency of all three strategies in the population, with the vertices representing scenarios where only one of the three strategies are present in the population. The color-coding of each point in $S_3$ represents the number of times, out of 1000 realizations, the population's composition was represented by the corresponding point in the simplex.

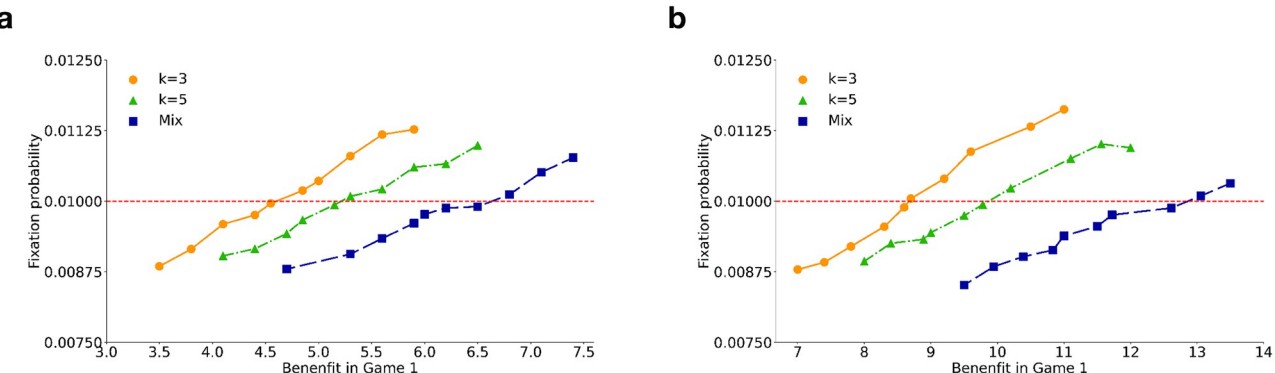

**Fig 5. Variation of fixation probability of the LGTFT strategy with benefit in game 1 for different degrees. (a)** $q = 0$, **(b)** $q = 0.3$. The fixation probability was calculated from $10^6$ trials. The dotted horizontal line shows the neutral fixation probability. The network was created using RRN topology. Other parameters used: $b_2 = 2$, $c_1 = c_2 = 1$, $w = 0.5$, $s = 0.01$, $N = 100$.

For weak selection strengths, both LGTFT and ALLD are abundant in the population with ALLD being more dominant when $b_1 = 1.5$ (Fig 6a and 6e) and LGTFT being more dominant when $b_1 = 4$ (Fig 6c and 6g) irrespective of the underlying population structure. This is also reflected in the cooperation rate which is larger in the latter case for both well-mixed (Fig 6c) and structured populations (Fig 6g). For large selection strengths, almost all players play the ALLD strategy in both well-mixed and structured populations when $b_1 = 1.5$ as indicated by the red circle in (Fig 6b and 6f). For $b_1 = 4$, ALLD remains dominant in well-mixed populations (indicated by the red circle in Fig 6d), while in structured populations, a mixed state of ALLC and LGTFT dominates, leading to a high cooperation rate ($C = 0.99$) in the population (Fig 6h). These findings are consistent with the behavior observed in our simulations carried out using the full strategy space at large $N$ (Fig 4d–4f), where the cooperation rates and

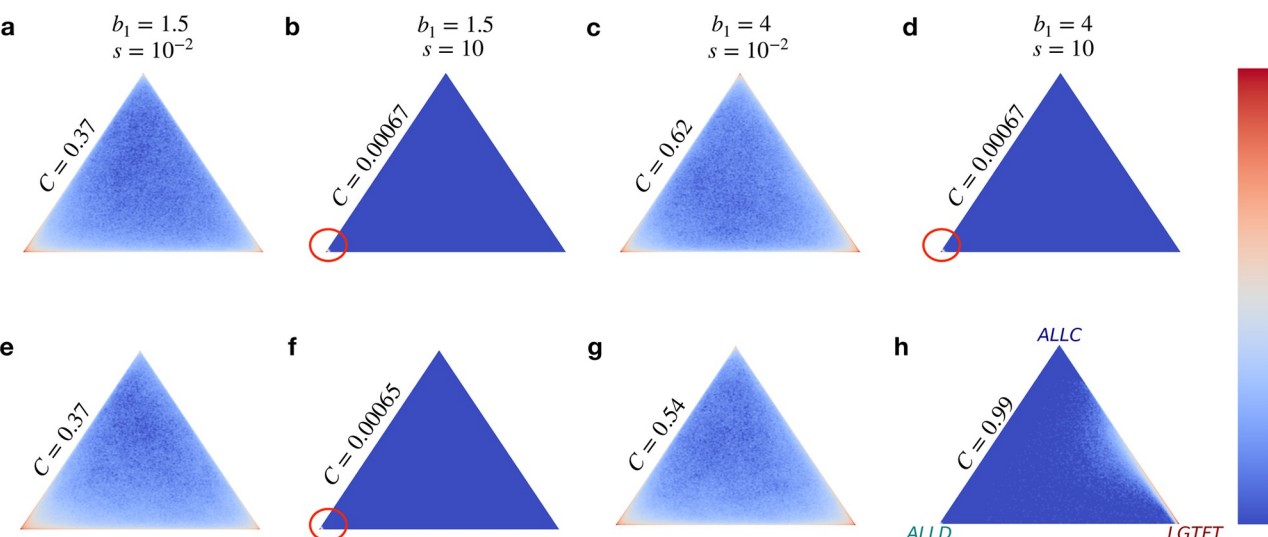

**Fig 6. Heat map showing likelihood of different population compositions in the reduced strategy system {ALLC, ALLD, LGTFT}.** Each point in the simplex represents the composition of the population averaged over the last $10^3$ time-steps out of a total simulation time of $10^5$ time-steps. The color of each point in the simplex denotes the number of times, out of $10^3$ independent simulations, the population appears with the specified abundance; with dark red being high and dark blue being low. (**a-d**) represent well-mixed and (**e-h**) represent structured population. The cooperation rate ($C$) is written above one edge of the simplex. Parameters: $N = 200$, $b_2 = 2$, $c_1 = c_2 = 1$, $\mu = 0.001$, $w = 1$. For LGTFT, $q = 0.3$.

abundance of full cooperation scenarios were found to be higher in multiplex-structured populations.

Our analysis with both an unconstrained strategy set as well as a reduced strategy set unambiguously confirm that a multiplex network structure facilitates the dominance of linked cooperative strategies (such as LGTFT) in multichannel games, resulting in high levels of cooperation in structured populations of all sizes. In contrast, selfish strategies like ALLD start dominating in well-mixed populations as the population size increases.

### 3.3 Evolution of cooperation in multiplex networks having identical topologies but distinct connections in each layer

So far, we have focused on multiplex networks with complete edge overlap between layers ($\mathcal{O} = 1$), which implies that each individual's neighbors are identical across different social contexts. However, in real world scenarios, individuals are typically members of multiple social networks with each network having a distinct set of connections i.e. a distinct network structure. An individual may have some neighbors who are common across all layers of the multiplex and others that are unique to a specific layer of the multiplex. Since linking of strategies across layers is possible only for those neighbors of the focal player that are common across all layers of the multiplex, this effect can be investigated by systematically varying the number of common neighbors of each member of the population. We do so using a two-layer multiplex network, where we could adjust the structural similarities between the layers. Specifically, we considered two layers of Random regular network, each consisting of 100 nodes. By tuning the average edge overlap $\mathcal{O}$ (Eq 2, for 2 layers $\mathcal{O}^{\alpha,\beta} = \mathcal{O}$) of the multiplex, we can control the degree of similarity between the layers. Our model with variable edge overlap extends the work of [55] by providing a general framework to explore the impact of linking strategies on the evolution of cooperation for any fractions of common neighbors across layers. This perspective enhances our understanding of the effectiveness of strategy linking in more realistic scenarios represented by variable degrees of edge overlap of the multiplex network. Note that results in this and following sections have been obtained using the independent strategy update scheme only.

Fig 7a shows the change in linked cooperation rate of population and Fig 7b represents the change in cooperation rate of population as the fraction of common neighbors across layers is varied. As expected, the linked cooperation rate ($\zeta_{CN}^{\alpha}$, Eq 11) calculated against common neighbors only, shows a monotonically increasing trend in both layers with increasing fraction of common neighbors. In contrast, the cooperation rate ($\zeta^{\alpha}$, Eq 12), calculated against all neighbors (both common and unique) shows an U-shaped nature with change in fraction of common neighbors. Fig 7b reveals that when individuals have a few common neighbors with whom strategies can be linked, strategy linking against common neighbors is not very effective in terms of boosting cooperation compared to the case without linking, i.e. when edge overlap $\mathcal{O} = 0$. Linking of strategies is detrimental to cooperation unless the degree of edge overlap is significantly high (i.e. $\mathcal{O} \gtrsim 0.7$).

When the benefit ($b_1$) in layer 1 is very low ($< b_2$), increasing the fraction of common neighbors has no effect on the linked cooperation rate (see Fig 8a). However, for sufficiently large $b_1$, increasing the fraction of common neighbors enhances the linked cooperation rate as expected since individuals with linked strategies are less prone to exploitation by selfish opponents in both games. On the other hand, the cooperation rate that considers both common and unique neighbors show an interesting behavior. When the benefit ($b_1$) in layer 1 is above a minimum threshold, increasing the fraction of common neighbors initially has a detrimental effect on the cooperation rate in layer 1 (see Fig 8b). However, further sustained increase in

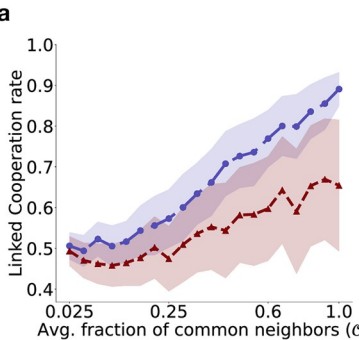
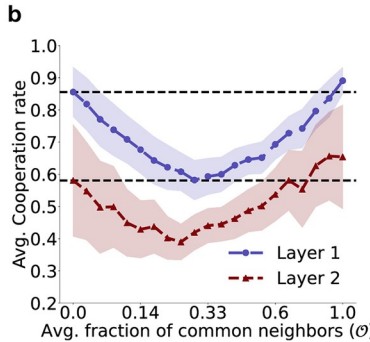
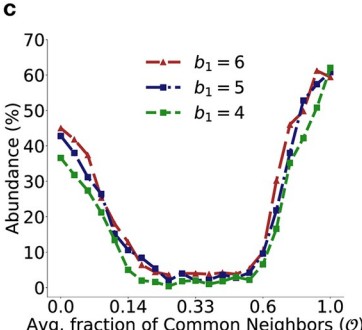

**Fig 7. Impact of changing fraction of common neighbors in a multiplex network having RRN topology. (a)** Change in the linked cooperation rate ($\zeta_{CN}^\alpha$), **(b)** cooperation rate ($\zeta^\alpha$) in both layers, and **(c)** abundance of full cooperation scenarios; averaged over multiple independent simulations; with change in fraction of common neighbors across layers when an individual uses multi-game linked (unlinked) reactive strategies while interacting with common (unique) neighbors in both layers. Each point corresponds to the mean of 500 independent simulations each of which ran for 60,000 time-steps. The averaging over trials was done at the end of 60,000 time-steps. The shaded region depicts one sigma variation from the mean. The dashed horizontal lines in panel **(b)** represent the cooperation rate in each layer (1 & 2), when the fraction of common neighbors across layers $\mathcal{O} = 0$ and are included as guides to the eye. Population size in each layer was $N = 100$, $k = 20$. All other parameters used are same as Fig 2.

the fraction of common neighbors eventually leads to an increases in the cooperation rate (see Fig 8b). This behavior generalizes the results depicted in Fig 7b, where a U-shaped nature of the cooperation rate was observed for a fixed value of $b_1$. An initial increase in the average fraction of common neighbors leads to a drop in cooperation rate in either or both layers of the multiplex as evident from the decrease in the abundance in full cooperation scenario (see Fig 7c) where $\zeta^\alpha \geq 0.8$ for $\alpha = 1, 2$. For $\mathcal{O} > 0.6$, the abundance of full cooperation scenario increases due to the increased advantage associated with strategy linking with a majority of neighbors, the cooperation rate also increases eventually reaching a maximum for complete overlap when $\mathcal{O} = 1$. These results indicate that strategy linking is not always effective in enhancing the cooperation rate in multiplex structured populations and a significant amount of edge overlap between the two layers of the multiplex is necessary for strategy linking to yield the benefits of enhanced cooperation.

To gain deeper insights into these results, we analyzed the strategies players employ against their common and unique neighbors as we varied the fraction of edge overlap ($\mathcal{O}$) between the two layers of the multiplex network (see S1 Fig). When edge overlap is zero or one, players on average use strategies similar to those observed in Fig 2d and 2h, reciprocating altruistic acts with high probability. However, for intermediate edge overlap, we observed (see S1(b) and S1(d) Fig) that the evolved strategies on the average are less likely to reciprocate a co-player's altruistic act in both games, leading to lower cooperation rates compared to $\mathcal{O} = 0$ or $\mathcal{O} = 1$ cases. This suggests that the degree of edge overlap can affect the nature of strategies that evolve with less altruistic strategies being employed against both common and unique neighbors for intermediate levels of edge overlap.

Fig 9 is a heat map showing the variation of the cooperation rate as the benefit of cooperation in both layers of the multiplex network is varied, for different degrees of edge overlap, corresponding to (a) $\mathcal{O} = 0$, (b) $\mathcal{O} = 0.5$, (c) $\mathcal{O} = 1$. A comparison of the three panels indicate that the cooperation rate exceeds 0.5 over a larger region of $b_1$-$b_2$ space for zero and complete edge overlaps of the two layers of the multiplex. For intermediate degrees of edge-overlap (such as $\mathcal{O} = 0.5$), increasing the benefits in layer $\alpha$ favours increased cooperative behavior in that layer due to a higher chance of layer $\alpha$ cooperation scenarios emerging at the end of the simulation for each trial.

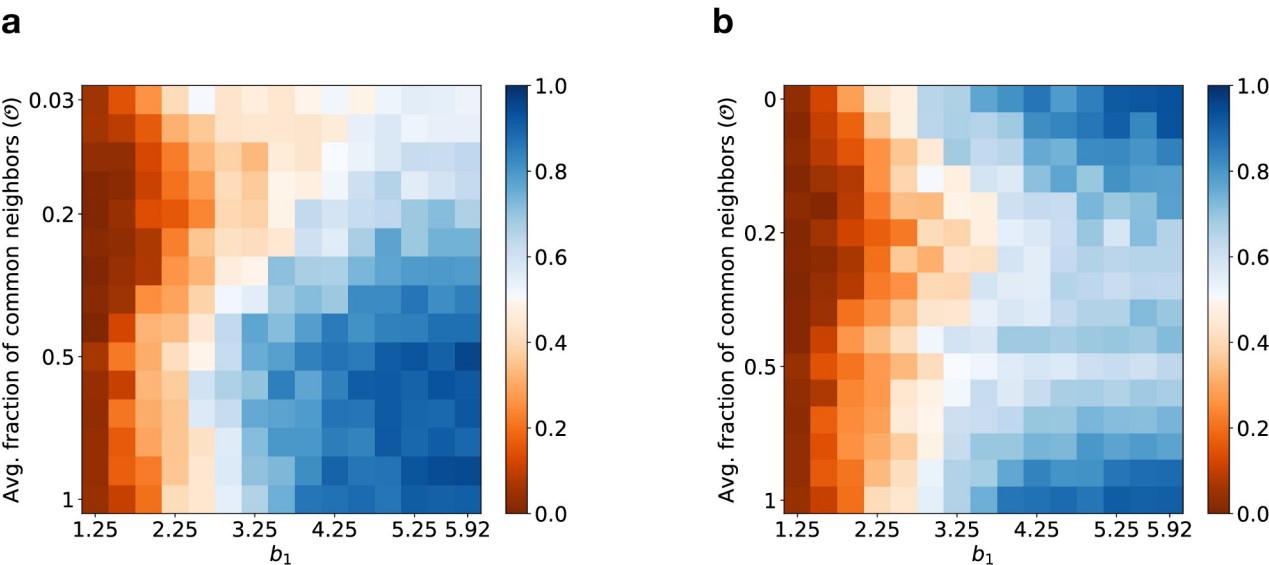

**Fig 8. Impact of changing fraction of common neighbors and benefit of cooperation in layer 1.** **(a)** linked cooperation rate ($\zeta^1_{CN}$) and **(b)** cooperation rate ($\zeta^1$) in layer 1. Each layer of the multiplex was created using RRN topology with degree $k = 15$. The value of each pixel corresponding to the respective types of cooperation rate was calculated at the end of the simulation that was run for 60,000 time-steps and averaged over 100 independent trials. The benefit in game 2 was kept fixed at $b_2 = 1.5$ and $c_1 = c_2 = 1$. All other parameters used are same as in Fig 7.

## 3.4 Incorporating imperfect memory as a cognitive constraint in repeated games

The success of conditionally cooperative strategies, such as GTFT in single repeated PD games and LGTFT in multiple repeated PD games depend on players having perfect memory of co-player's past actions and correctly implementing their intended actions in the current round. While the impact of implementation error in repeated games is well-studied [6, 7, 18, 63, 64], most of these studies assume perfect memory for all players. Everyday experience and prior experiments [65], [66] indicate that people involved in interactions across multiple contexts often confuse past outcomes [67] and misremember previous actions. Previous theoretical studies on the evolution of cooperation have ignored the cognitive constraints of agents in social interactions. In the following, we study the impact of imperfect memory on the

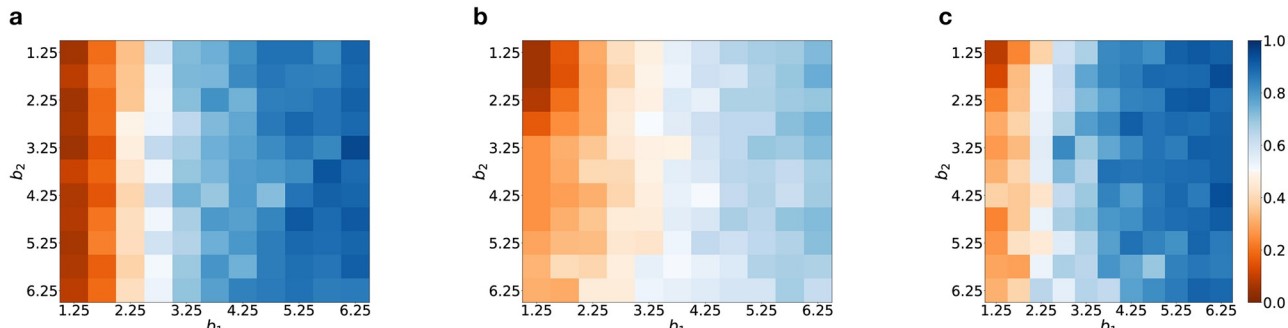

**Fig 9. Heatmap depicting change in layer 1 cooperation rate with varying benefits associated with cooperation in both layers.** The three panels correspond to cases where the degree of edge overlap is **(a)** $\mathcal{O} = 0$, **(b)** $\mathcal{O} = 0.5$, **(c)** $\mathcal{O} = 1$. The multiplex network has RRN topology in each layer. Other parameters are same as those used in Fig 7.

evolution of cooperation in multiple repeated games on multiplex network. For detailed derivations and discussions, see Sec.S2.5 of S1 Text.

To account for imperfect memory, we assume players misremember game 1 and game 2 outcomes with probabilities $\varepsilon_{IM}^1$ and $\varepsilon_{IM}^2$, respectively. If a player is subject to imperfect memory in game 1, they will recall the opponent's previous cooperation ($C$) as defection ($D$) and vice versa in game 1, while correctly recalling the co-player's action ($a_2 \in \{C, D\}$) in game 2. If an error occurs in both games, a co-player's cooperation in both games ($CC$) will be misremembered as defection ($DD$) in both games. In the former case of error in game 1 only, a player cooperates with probability $p_D^1$ instead of $p_C^1$ in game 1 against a unique neighbor in layer 1. For a common neighbor across layers, the player cooperates with $p_{D,a_2}^1$ in game 1 and $p_{D,a_2}^2$ in game 2, instead of $p_{C,a_2}^1$ and $p_{C,a_2}^2$, respectively. In the latter case of error in both games, a player cooperates with probabilities $p_D^1$ and $p_D^2$ in game 1 and game 2, respectively, instead of $p_C^1$ and $p_C^2$ against unique neighbors in layers 1 and 2. Against common neighbors, cooperation probabilities shift to $p_{DD}^1$ and $p_{DD}^2$ in game 1 and game 2, respectively, from $p_{CC}^1$ and $p_{CC}^2$. Here, we aim to understand the effects of imperfect memory in two cases: **(i)** when individuals use unlinked strategies against all neighbors ($\mathcal{O} = 0$), and **(ii)** when individuals use linked strategies against all neighbors across all layers ($\mathcal{O} = 1$). For unique neighbors, an error in one game does not affect the other, as each game is treated independently. In contrast, for common neighbors, errors in one game can significantly impact the other due to strategy linking across games. In line with this intuition, we varied error probability ($\varepsilon_{IM}^1$) in game 1, while keeping perfect memory for game 2 ($\varepsilon_{IM}^2 = 0$). Fig 10a shows that for unlinked strategies ($\mathcal{O} = 0$), the cooperation rate in game 2 remains around 85%, while in game 1, it declines rapidly from 94% to 0% with increasing error probability. Fig 10b illustrates the variation in linked cooperation rates ($\mathcal{O} = 1$) in both games. Despite no errors in game 2, a moderate decrease in cooperation rate from 83% to 61% is observed due to the linking of strategies. Cooperation rate in game 1 becomes vanishingly small with increasing error probability. However, unlinked cooperation rate in game 1 decreases faster than the linked cooperation rate. To understand how errors in both games impact cooperation, we assumed error probability in both games are equal $\varepsilon_{IM}^1 = \varepsilon_{IM}^2 = \varepsilon_{IM}$ and we varied $\varepsilon_{IM}$ continuously for both cases of $\mathcal{O} = 0$ (Fig 10c) and 1 (Fig 10d). Consistent with previous experiments, we have seen that such errors have a significant negative impact on cooperation for scenarios with only unique and only common neighbors. For moderately high error probability ($\gtrsim 0.3$) cooperation quickly diminishes in repeated games. Our results with imperfect memory underscores the importance of incorporating cognitive constraints in future research on repeated games.

## 3.5 Empirical multiplex network

We examined two real-world communities each interacting in multiple social contexts. We analyzed a two-layered, undirected multiplex network representing social (meeting for lunch) and working (research collaboration) relationships within the Aarhus University CS department [68], as well as business relationships and marriage alliances among Florentine families in the Renaissance [69] (see Sec.S4 of S1 Text). The population size of each network ranged from $N = 16$ to $N = 61$. Across these communities, the average fraction of edge overlap varied from $\mathcal{O} = 0.296$ to $\mathcal{O} = 0.34$. Such networks deviate from the RRN networks studied in earlier sections since distinct nodes in each layer can have different degrees and different levels of edge overlap with corresponding nodes in the other layer. Fig 11a and 11b shows visualisations of such multiplex networks. In our analysis, we assumed that if an individual shares a common interacting partner across all contexts (online/offline interactions, business partnerships/marriage ties), they will employ a linked strategy when interacting with that partner. We wish to

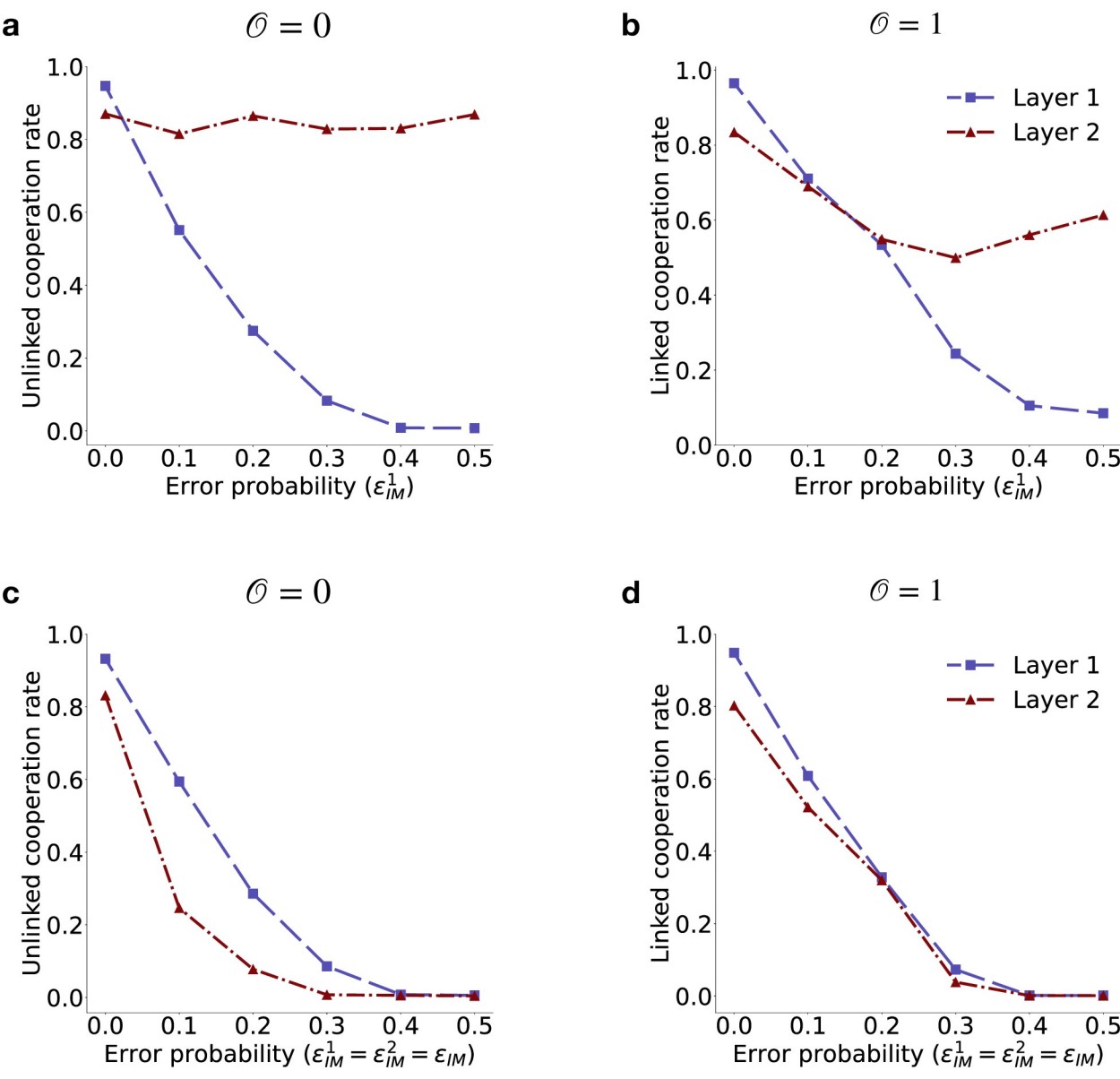

**Fig 10. Impact of imperfect memory.** The population's cooperation rate when all individuals use either (**a, c**) unlinked strategy ($\mathcal{O} = 0$) or (**b, d**) linked strategy ($\mathcal{O} = 1$). In panels **a-b**, we varied the perception error in game 1 while game 2 was assumed to be error-free. In panels **c-d**, we varied the error in both games simultaneously. Incorporation of imperfect memory shows the negative impact of linking, as errors in game 1 reduce cooperation in game 2 (**b**), whereas unlinked cooperation in game 2 remains unaffected by increasing errors in game 1 (**a**). However, as errors increase in both games, cooperation diminishes rapidly for both unlinked (**c**) and linked (**d**) strategies. Each point represents the population's cooperation rate, averaged over the last 1,000 time steps and 100 independent realizations. Each simulation of evolutionary dynamics ran for $10^5$ time-steps of independent strategy update. Both layers of multiplex network follow an RRN topology with nodes $N = 100$ and a degree, $k = 4$. Other parameters are the same as in Fig 2.

understand how cooperation evolves in these real-world multiplex networks where individuals repeatedly engage in several interactions with various partners across multiple contexts.

We randomly selected a layer of the multiplex network and assumed the benefit of cooperation in this layer (blue layer, Fig 11) is higher than in the other layer (brown layer). As expected, a higher level of cooperation evolved in the layer with greater benefit. Fig 11c and 11d illustrates the temporal evolution of cooperation in two layers of the multiplex network of

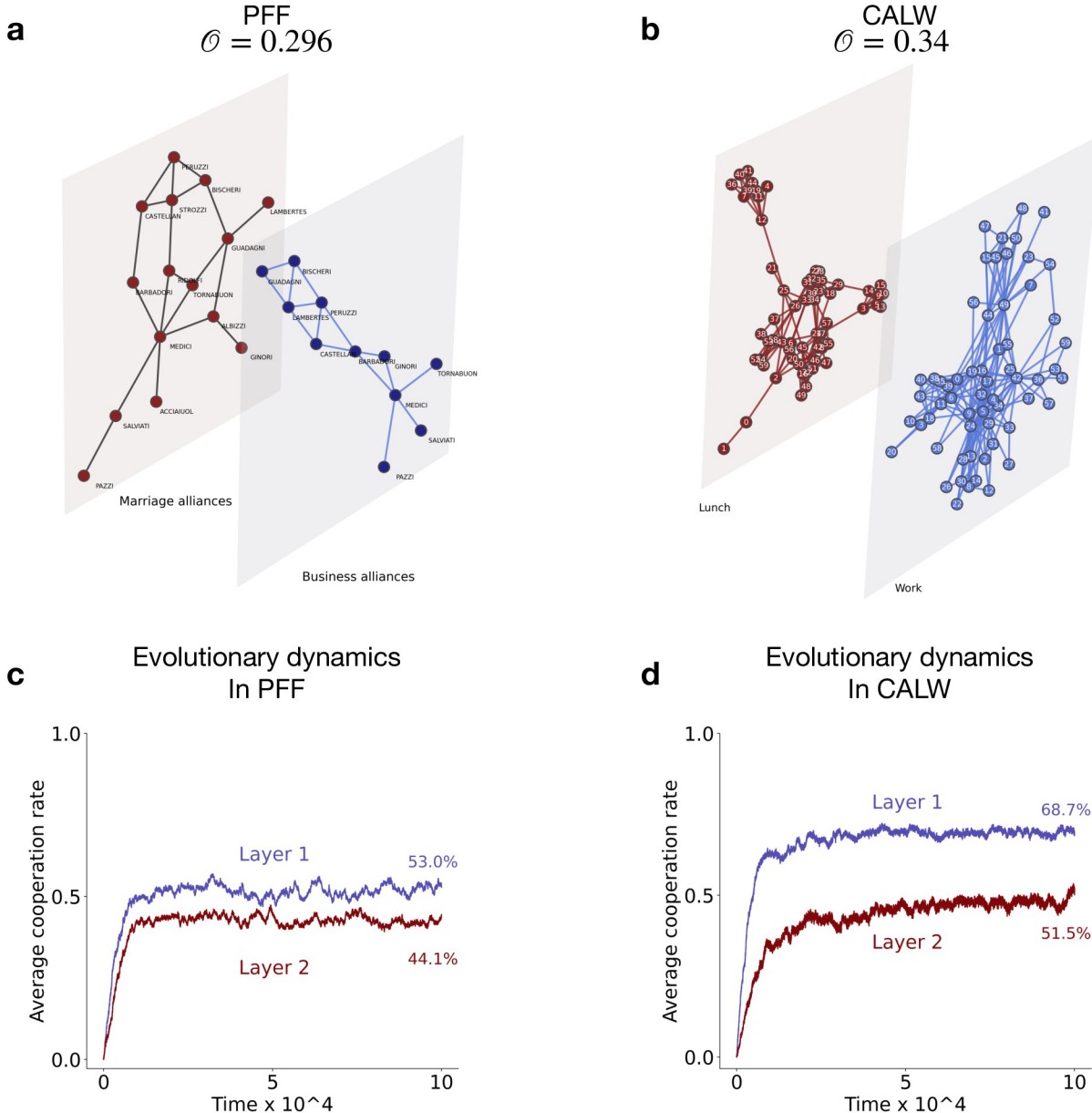

**Fig 11. Evolution of cooperation in two-layered, undirected real-world multiplex networks**: Visualizations of **(a)** the marriage alliance and business relationships of $N = 16$ Florentine families during the Renaissance (PFF) as a 2-layer multiplex network structure of Florentine families; **(b)** different offline relationships (eating lunch together and work-related collaboration) between $N = 61$ employees of the CS department in the Aarhus University (CALW). **(c)** & **(d)** shows the temporal evolution of populations' cooperation rate in each layer. Each point is an average of 100 independent simulations using independent strategy update. One layer (blue) has a higher benefit of cooperation than the other (brown). Parameters used are the same as in Fig 2.

Florentine families and CS-Aarhus employees who share lunch and also collaborate on common work projects respectively. We observed that higher cooperation levels in both layers evolved in multiplex networks where the average edge overlap ($\mathcal{O}$) is higher. This finding aligns with our results obtained in Sec. 3.3 using a RRN topology.

## 3.6 Other games

Even though our main focus in this paper was on the donation game, we also obtained results showing the contrasting impact of linking strategies in the SD [70] and SH [71] games (see S2 and S3 Figs) when compared to the donation game. We find that when games in both layers are SD games, linking enhances the cooperation rate in the lower benefit game. However, the distribution of the emergent cooperation scenarios is quite different. Even though the abundance of the full cooperation scenario is significantly higher when strategies are linked, it is no longer the dominant scenario. The most abundant scenario corresponds to the case where the cooperation rate in both layers is less than 80% but greater than 20% (see S2 Fig). Moreover, a structured population does not confer any additional advantage in terms of enhancing cooperation rate, as has also been reported elsewhere [70]. In the sculling game, which is a version of the SH game, the differences are even more striking, with linking being counterproductive to the spread of cooperation in layer 2, where the cooperation rate dropped from approximately 77% in the unlinked case to around 19% in the linked case. On the contrary, the very high cooperation rate observed in layer 1 for the SH game can be attributed to the fact that our parameter choice of the game 1 payoff matrix makes cooperation risk dominant. However, this is not true for game 2 where defection is risk dominant, as a result of which cooperation levels are extremely low in that game. The abundance of the full cooperation scenario in the linked case decreases with only layer 1 cooperation scenario being the most abundant (see S3 Fig).

## 4 Discussion

It is necessary to account for complex social interactions, that can occur across multiple domains, while attempting to decipher the underlying mechanisms that promote and sustain cooperation in human societies. It is in this spirit that we extended the framework of multichannel games to multiplex networks. Moreover, such a framework allows us to understand how a strategic decision in one layer of the multiplex can affect decision in another layer and how strategy linking across layers of a multiplex affect the spread of cooperation in different layers. The efficacy of strategy linking in a multiplex network-structured population depends on a wide variety of factors like population size, degree of each network layer, extent of overlap between different network layers, perception errors and the strategy update rule. When the topology as well as the structure of the different layers of the multiplex are identical, strategy linking leads to greater abundance of the full cooperation scenario in PD games and higher cooperation rate in all layers of the multiplex compared to the mixed-population case, especially for large population sizes when the strategy of all layers are simultaneously updated. Moreover, strategy linking is found to induce high cooperation rates (60% or higher) in *both* layers only for large structural overlaps ($\mathcal{O} \geq 60\%$) between the two layers. When the number of common neighbors is small, the benefits of linking strategies across layers do not accrue as much. The lack of coherence between linked and unlinked strategies ensures that the cooperation rate in both layers decreases with the initial increases in the fraction of common neighbors. However, when the fraction of common neighbors crosses a certain threshold, the ability of individuals to use linked strategies while playing with most of their neighbors enhance the cooperation rate in both layers. This is also manifest through the increase in abundance of full cooperation scenarios. Linking strategies also makes the system more susceptible to the negative consequences of errors in decision-making. Perception errors in just one game can lower the cooperation rate not only in that game but also in the game where there are no such errors.

Our work also opens up the possibility of using agent-based models to explore different types of strategy linking mechanisms across the layers of the multiplex network and their

consequences on the evolution of cooperation. For instance, individuals may be more likely to cooperate in one layer if they are present in a cooperative environment in another layer. This underscores the importance of the strategy environment [72] in one layer affecting cooperative behavior in other layers of the multiplex. Moreover, the issue of how the multiplex network structure affects the spread of cooperative behavior in other social dilemmas like the public goods game, remains a pertinent question that is also open to future exploration. Our work suggests that multiplex networks provide a rich tapestry for emergent patterns of cooperative behavior that would not be seen in well-mixed populations and single layered networks.

## Supporting information

**S1 Fig. Average strategy employed for different fraction of common neighbours.** Strategies players use (on an average) against their unique neighbors (panel **a-b**) and common neighbors (panel **c-d**) when the average fraction of common neighbors across two layers of the multiplex network are (**a**) 0, (**b, d**) 0.33, and (**c**) 1 respectively. $\mathcal{O} = 0.33$ indicates each individual has $O_i^{CN} = k/2 = 10$ common neighbors, where $k = 20$ is the degree of each node of the RRN in both layers of the multiplex network with $N = 100$ nodes per layer. Each bar represents the population's average of $p_{a_1,a_2}^{\alpha}$ ($p_{a_\alpha}^{\alpha}$) against common (unique) neighbors in that respective layer, averaged over 100 independent realizations using independent strategy update, whereas dots represent 50 randomly sampled realizations of the simulation. Other parameter values used are the same as in Fig 7.
(PDF)

**S2 Fig. The effect of linkage and network connectivity on the Snowdrift game.** The Snowdrift game [70] is a version of the Chicken game, where defection is not an equilibrium of the game. In this game, it is always best to cooperate if the other individual defects and vice-versa. Using unlinked (**a-d**) and linked (**e-h**) strategies, we simulate the dynamics when players simultaneously engage in a game with larger benefit of cooperation (game 1, blue), and a game with relatively lower benefit of cooperation (game 2, brown) in layer 1 and layer 2 of a multiplex network respectively. (**a,e**) shows the variation of the cooperation rate with degree ($k^1 = k^2 = k$); (**b,f**) the time evolution of the cooperation rate for $k = 4$; (**c,g**) relative abundance of different cooperation scenarios across multiple simulations for $k = 4$; (**d,h**) the average strategy used for $k = 4$; when all individuals use unlinked and linked strategies respectively. The results were obtained by averaging over 100 simulations using the independent strategy update rule and RRN topology for each layer of the multiplex network. For Snowdrift games considered here, $R_1 = b_1 - \frac{c}{2}$, $S_1 = b_1 - c$, $T_1 = b_1$, $P_1 = 0$, and $R_2 = b_2 - \frac{c}{2}$, $S_2 = b_2 - c$, $T_2 = b_2$, $P_2 = 0$. Other parameters used: $N = 100$, $b_1 = 2.5$, $b_2 = 1.2$, $c = 1$, $\mu = 0.001$, $w = 1$, and $s = 2$.
(PDF)

**S3 Fig. The effect of linkage and network connectivity on the Sculling game.** The Sculling game [71] is a variant of the Stag-Hunt game in which the benefits of cooperation depends non-linearly on the number of cooperators. For the selected parameters, this game falls within the coordination class, where mutual cooperation is an equilibrium even if the game is repeated only once. Using unlinked (**a-d**) and linked (**e-h**) strategies, we simulate the dynamics when players simultaneously engage in a game with larger benefit of cooperation (game 1, blue), and a game with relatively lower benefit of cooperation (game 2, brown) in layer 1 and layer 2 of a multiplex network respectively. (**a,e**) shows the variation of the cooperation rate with degree ($k^1 = k^2 = k$); (**b,f**) the time evolution of the cooperation rate for $k = 4$; (**c,g**) relative abundance of different cooperation scenarios across multiple simulations for $k = 4$; (**d,h**) the average strategy used for $k = 4$; when all individuals use unlinked and linked strategies

respectively. The results were obtained by averaging over 100 simulations using the independent strategy update rule and RRN topology for each layer of the multiplex network. For Sculling games considered here, $R_1 = \frac{4}{3}b_1 - c$, $S_1 = \frac{b_1}{3} - c$, $T_1 = \frac{b_1}{3}$, $P_1 = 0$, and $R_2 = \frac{4}{3}b_2 - c$, $S_2 = \frac{b_2}{3} - c$, $T_2 = \frac{b_2}{3}$, $P_2 = 0$. Other parameters used: $N = 100$, $b_1 = 2.5$, $b_2 = 1.2$, $c = 1$, $\mu = 0.001$, $w = 1$, and $s = 2$.
(PDF)

**S1 Text. Supporting information text.**
(PDF)

## Acknowledgments

AB acknowledges the support provided by the Kepler Computing facility, maintained by the Department of Physical Sciences, IISER Kolkata. S.S. acknowledges partial support provided by a MATRICS grant no. (MTR/2020/000446, 2020-2023), given by SERB, India.

## Author Contributions

**Conceptualization:** Amit Basak, Supratim Sengupta.

**Data curation:** Amit Basak.

**Formal analysis:** Amit Basak, Supratim Sengupta.

**Funding acquisition:** Supratim Sengupta.

**Investigation:** Amit Basak.

**Methodology:** Amit Basak, Supratim Sengupta.

**Project administration:** Supratim Sengupta.

**Resources:** Supratim Sengupta.

**Software:** Amit Basak, Supratim Sengupta.

**Validation:** Amit Basak.

**Writing – original draft:** Amit Basak, Supratim Sengupta.

**Writing – review & editing:** Amit Basak, Supratim Sengupta.

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
