## [Decision Letter · Decision Letter 0]

21 Oct 2024

Dear Dr Sengupta,

Thank you very much for submitting your manuscript "Evolution of cooperation in multichannel games on multiplex networks" for consideration at PLOS Computational Biology.

As with all papers reviewed by the journal, your manuscript was reviewed by members of the editorial board and by several independent reviewers. In light of the reviews (below this email), we would like to invite the resubmission of a significantly-revised version that takes into account the reviewers' comments. Please, pay special attention to provide a thorough comparison with previous literature, as we aim to publish manuscript that are highly innovative in their field.

We cannot make any decision about publication until we have seen the revised manuscript and your response to the reviewers' comments. Your revised manuscript is also likely to be sent to reviewers for further evaluation.

Sincerely,

Yamir Moreno

Academic Editor

PLOS Computational Biology

Tobias Bollenbach

Section Editor

PLOS Computational Biology

Reviewer's Responses to Questions

**Comments to the Authors:**

Reviewer #1: This work successfully establishes a model of new multiplex networks (Layers-1 and 2), where each plays different PD in terms of dilemma strength in each layer. One novel point in the authors’ modeling is that, depending on whether a focal edge is unique in one of the networks or common (the edge exists in both layers), the focal player’s offer (C or D), i.e., updating procedure is different. As Fig. 1 intelligibly explains, an offer on unique edge stochastically references to what the game opponent in the edge (single memory strategy), while, in the case of common edge, this reflection affects on both offers on the focal agent. I guess that no one has ever studied such a model. In this sense, I positively evaluate this work.

The authors mainly delivered MAS results in visuals.

For example, Figs. 2 & 3 confirm that emerged cooperation in layer 1 is more enhanced than that of layer 2. Honestly speaking, to me, it’s not specifically surprising, simply because the dilemma weakness (that is inverse to the dilemma strength, which is mentioned as below) at layer 1 is more than that at layer 2 due to b_1 = 5 > b_2 – 3 while c_1 = c_2 =1.

One suggestion relevant to the point above is that the authors should mention about the concept of universal dilemma strength, quantified by the Chicken-type dilemma; Dg’ := (T - R) / (R – P) and Stag Hunt-type dilemma; Dr’ := (P – S) / (R – P). And they should note that the Donor & Recipient (D & R) game (the authors called the donation game) is special because of the same amount of Dg’ = Dr’ = c / (b – c), which is appropriate for a typical Prisoner’s Dilemma game. They are suggested to add those points in the introduction or model depiction section by referring to relevant literatures; (i) Relationship between dilemma occurrence and the existence of a weakly dominant strategy in a two-player symmetric game, BioSystems 90(1), 105-114, 2007, (ii) Universal scaling for the dilemma strength in evolutionary games, Physics of Life Reviews 14, 1-30, 2015, (iii) Scaling the phase- planes of social dilemma strengths shows game-class changes in the five rules governing the evolution of cooperation, Royal Society Open Science, 181085, 2018, (iv) Sociophysics Approach to Epidemics, Springer, 2021.

Reviewer #2: The manuscript presents a model where individuals engage in Prisoner’s Dilemma (PD) games within a multiplex network, with various layers representing different interaction contexts (different PD games). The main focus is on how cooperation evolves based on the network structure, specifically the edge overlap between layers, and how imperfections in memory influence strategies. This is indeed an interesting topic, as cooperation in complex networks remains a relevant issue in evolutionary game theory.

The use of real-world data, such as the interactions among Florentine families and employees at Aarhus University, adds significant value to the study. It effectively demonstrates how theoretical models can be applied to real-life situations, enriching the analysis of cooperation dynamics.

However, there are a few areas where the paper could be enhanced. While the approach is relevant, there are numerous studies addressing cooperation in multilayer and multiplex networks. The paper would benefit from a clearer articulation of what distinguishes this work from existing research. A more explicit comparison with previous models would help contextualize the contributions and highlight their uniqueness. Without this context, even readers and researchers interested in this field may find it challenging to assess the manuscript's originality; thus, contextualizing the results with prior research is essential.

**Have the authors made all data and (if applicable) computational code underlying the findings in their manuscript fully available?**

Reviewer #1: Yes

Reviewer #2: Yes

PLOS authors have the option to publish the peer review history of their article (what does this mean?). If published, this will include your full peer review and any attached files.

Reviewer #1: No

Reviewer #2: No
---

## [Decision Letter · Decision Letter 1]

27 Nov 2024

Dear Dr Sengupta,

We are pleased to inform you that your manuscript 'Evolution of cooperation in multichannel games on multiplex networks' has been provisionally accepted for publication in PLOS Computational Biology.

Best regards,

Yamir Moreno

Academic Editor

PLOS Computational Biology

Tobias Bollenbach

Section Editor

PLOS Computational Biology

Feilim Mac Gabhann

Editor-in-Chief

PLOS Computational Biology

Jason Papin

Editor-in-Chief

PLOS Computational Biology

Reviewer's Responses to Questions

**Comments to the Authors:**

Reviewer #1: The revised MS is good rnough for a publication on the journal.

Reviewer #2: In my opinion, all of the referees' suggestions have been adequately addressed. Therefore, I recommend the manuscript for publication in PLOS Computational Biology.

**Have the authors made all data and (if applicable) computational code underlying the findings in their manuscript fully available?**

Reviewer #1: None

Reviewer #2: None

PLOS authors have the option to publish the peer review history of their article (what does this mean?). If published, this will include your full peer review and any attached files.

Reviewer #1: No

Reviewer #2: No

---

## [Editor Report · Acceptance letter]

12 Dec 2024

PCOMPBIOL-D-24-01613R1 

Evolution of cooperation in multichannel games on multiplex networks

Dear Dr Sengupta,

I am pleased to inform you that your manuscript has been formally accepted for publication in PLOS Computational Biology. Your manuscript is now with our production department and you will be notified of the publication date in due course.

With kind regards,

Anita Estes
